# Analysis of Stresses in a Tapered Roller Bearing Using Three-Dimensional Photoelasticity and Stereolithography

**DOI:** 10.3390/ma12203427

**Published:** 2019-10-20

**Authors:** Filipe Gomes Vieira, Alexandre S. Scari, Pedro Américo Almeida Magalhães Júnior, Jordana S. R. Martins, Cristina Almeida Magalhães

**Affiliations:** 1Department of Mechanical Engineering, Pontifícia Universidade Católica de Minas Gerais, Av. Dom José Gaspar, 500, Coração Eucarístico, Belo Horizonte, Minas Gerais 30535-901, Brazil; filipevieira_90@hotmail.com (F.G.V.); martinsjsr@gmail.com (J.S.R.M.); 2Department of Mechanical and Production Engineering, Universidade Federal de Juiz de Fora, R. José Lourenço Kelmer, s/n, São Pedro, Juiz de Fora, Minas Gerais 36036-900, Brazil; filipe.gomes@sga.pucminas.br; 3Department of Mechanical Engineering, Centro Federal de Educação Tecnológica de Minas Gerais, Av. Amazonas, 7675, Nova Gameleira, Belo Horizonte, Minas Gerais 30510-000, Brazil; paamj@oi.com.br

**Keywords:** photoelasticity, stereolithography, stress analysis

## Abstract

Digital photoelasticity is an important segment of optical metrology for stress analysis by digital photographic images. Advances in digital image processing, data acquisition, standard recognition and data storage allow the utilization of computer-aided techniques in the automation and improvement of the digital photoelastic technique. The objective of this study is to develop new techniques using 3D rapid prototyping with transparent resins in digital photoelasticity. Some innovations are proposed (e.g., a tapered roller bearing built with 3D rapid prototyping with transparent resin and the final assembly with the specimens prototyped separately). A metrology study is carried out with the new techniques developed.

## 1. Introduction

Experimental stress analysis techniques were widely used during prototype design, but as data processing and storage on computers advanced, they were replaced by numerical techniques. However, experimental techniques are still important in confirming the results obtained by experimental techniques providing more reliability and safety in prototype design.

Numerical methods, such as the finite element method, are widely used in the analysis of stresses, especially in complex structures for which there are no exact/analytical solutions [1,2,3,4,5]. However, numerical models need to be validated, for which there are two options: comparison with the exact/analytical solution (when available) and/or comparison with experimental tests [6,7,8,9]. Prototypes usually have a high cost and testing with them can take time beyond what is available for product development. Photoelasticity is then considered as a viable experimental alternative [10,11,12,13,14,15].

Rapid prototyping is a technique that is growing every day and is very useful in product development for allowing the visualization of the component and simulating the assembly, while still in the design phase [16,17,18,19,20,21]. The analysis of stresses by finite elements, widely used in the initial phase of product design, requires validation of the models, both experimentally and analytically (when possible). Therefore, applying the digital photoelastic technique to components obtained with rapid prototyping is very useful for product engineers since small 3D printers are already accessible in the same way as a common office multifunction printer [22,23,24,25,26]. For this, it is necessary that the resin be transparent and birefringent.

In this research, the bearing resin model is identical to the original device made of steel and no part was removed from the bearing to simplify experiments, which is common when the analyzed component has a complex shape. Therefore, there is no change in geometry that can influence the results. Research on three-dimensional photoelasticity and photoelastic models produced by stereolithography is rare. Thus, this research seeks to contribute to the development of three-dimensional photoelastic analysis by showing that producing models by stereolithography may be the only viable alternative for some components.

The objective of this work is to use rapid prototyping via three-dimensional printing with birefringent transparent resin in an innovative way to separately print each component of a tapered roller bearing and assemble it. Futhermore, we apply stress freezing method to measure stresses through photoelasticity.

## 2. Stereolithography and Photoelasticity

Stereolithography (SL) uses a photocurable liquid resin (based on acrylates and epoxy), and the cure is usually obtained by an ultraviolet (UV) laser. This resin is inserted into a container containing a layered platform that moves downwardly after each layer is constructed. An optical assembly moves the laser beam, which reproduces the 2D geometry obtained by slicing the part from the CAD system and filling the corresponding layer on the surface of the container with the photocurable resin. When exposed to the laser beam, the resin polymerizes and becomes solid, generating a layer smaller than 0.5 mm thick. The diameter of the laser is usually approximately 0.25 mm. For applications requiring high resolution, the diameter is reduced to 0.075 mm.

The most effective and commonly used method in three-dimensional photoelasticity is the stress freezing method, where the deformations and the associated optical responses of the photoelastic model under tension are locked on a molecular scale. The photoelastic model can then be sliced and analyzed to obtain the stresses within the part. Furthermore, the thickness of the slice should be sufficiently thin relative to the size of the photoelastic model to ensure that the stresses do not change in either magnitude or direction through the thickness of the slice [3,5,26].

This method is based on the diphasic behavior of various polymeric materials when heated, as these are formed by long chains (primary and mostly secondary) of hydrocarbon molecules. As the photoelastic model warms up, the chains of secondary bonds break, and the deformations occurring in the primary bonds are elastic. By cooling this photoelastic model under tension to room temperature, the secondary bonds again form between the elastically deformed primary bonds, and thus, the primary bonds are locked in their deformed positions [7,10,13,23].

In a tapered roller bearing, the inner and outer rings have different widths and contact angles. This causes a resultant force on the conical rollers, which compresses them against the rear face of the inner ring flange. Therefore, in applications requiring high speed, this bearing must be properly cooled and lubricated.

The contact angles allow the bearing to support axial loads. The cage clearance, on the other hand, influences the contact angles, the stresses, the deformations and the resistance to fatigue. For tapered roller bearings, in the absence of load, there is linear contact. However, when the bearing is subjected to loading, the linear contact becomes rectangular.

## 3. Methodology

The method of rapid prototyping chosen for this study was stereolithography because it presents good precision and one of the best surface qualities, in addition to high transparency. It is also noted that the VisiJet M3 Crystal resin of the MJM process was tested. However, this sample was very rigid and not transparent for photoelastic applications, making the appearance of the fringes difficult.

To define the most suitable transparent epoxy resin to be used in the stereolithography process and subsequent photoelastic analysis, two types were tested: Stereolithography (SLA)-ABS and SLA-Clear. To do so, a sample of each resin was made. The sample made of SLA-ABS resin proved to be very rigid and not transparent for photoelastic applications because it makes the appearance of the fringes difficult. The SLA-Clear resin sample was adequate for both flexibility and transparency. Its mechanical properties (given by producer) are shown in Table 1.

After defining the most appropriate resin, a sample of the tapered roller bearing was made in stereolithography (SLA). It is worth mentioning that each bearing component was printed separately as shown in Figure 1a and assembled in the sequence, as shown in Figure 1b,c. This means that the taper rollers are not attached to the cage or to the inner and outer rings. Thus, this prototype printed in 3D has the same degrees of freedom as the original bearing made with steel. Table 2 shows the main dimensions of the bearing studied.

Figure 2 presents the prototyped model in stereolithography positioned on a steel plate and subjected to a load of 1.3 kgf acting in compression. This model had its temperature elevated up to 58 °C (a step change with a duration of one hour). This temperature was maintained for one hour, and then the temperature was set to 40 °C and the cooling took place inside the oven. After freezing the stresses, this photoelastic model was sliced and positioned on the polariscope before and after slicing to obtain the digital photographs for later analysis.

The original bearing is made of ASTM 52100 steel, which has Poisson’s ratio ν_p_ = 0.3 and elastic modulus E_p_ = 203.4 GPa. These values were used in finite element analysis. The epoxy resin SLA-Clear has Poisson’s ratio ν_m_ = 0.4 and elastic modulus E_m_ = 2.90 GPa. In this work, we consider the application of this bearing in manual gearboxes in first gear, in this case, the bearing is subjected to axial effort of 2.04 kN. Equation (1) was used to calculate the equivalent load that was applied to the resin model. The axial load of F_m_ = 12.8 N ≈ 1.3kgf was applied to the resin model during the stress freezing method.

Equation (1) allows comparing the results obtained in the finite element analysis (FEA) with the experimental results. In Equation (1), σ = σ_p_/σ_m_ and F = F_p_/F_m_, σ refers to the ratio between results for FEA and photoelasticity, F gives the ideal value for σ and E refers to the difference between numerical and experimental results, the subscript m refers to the resin model and the subscript p refers to the steel bearing. The Equation (1) consider that the steel bearing has the same dimensions and Poisson’s ratio as the resin bearing.
(1)$$E=\frac{\left|\sigma -F\right|}{F}$$

The finite element models were generated from the existing 3D geometric model of the bearing. The mesh was generated with linear tetrahedral and hexahedral elements. The complete bearing model has the following characteristics:(a)elements used: C3D4 (linear tetrahedral) for the cage and tapered rollers (due to their geometric complexities), and C3D8 (linear hexahedral) for the outer and inner rings,(b)average element size: 0.5 mm,(c)number of nodes: 207,136,(d)number of elements: 360,230,(e)type of analysis: nonlinear static.

To analyze the model, the axial load due to the maximum torque in first gear was applied. For even distribution of this load, rigid elements were created, distributed in 360º and connected to the bearing by the inner ring. The union of these rigid elements occurs at the bearing centerline, where the axial load mentioned above was applied.

## 4. Results and Discussion

Figure 3 shows the tapered roller bearing model in stereolithography, where it is possible to visualize the largest diameter of all 18 tapered rollers. Table 3 shows the results for each part of the bearing in the photoelastic analysis.

Figure 4, Figure 5 and Figure 6 and Table 4 show the results of the two principal stresses for the bearing. For the first principal stress, the highest tensile stress occurred on the rear face of the outer ring, with a value equal to 334.7 MPa, as shown in Figure 4a. The tapered rollers, according to Figure 5a, preferably work under compression since the maximum value obtained for the first principal stress was 103.1 MPa, located on its rear face. In contrast, the inner ring had 133.2 MPa of traction on the rear face of its crimp, as shown in Figure 6a. The second principal stress showed −246.3 MPa for the outer ring, −693.4 MPa for the tapered rollers, and −292.7 MPa for the inner ring. These results are shown in Figure 4b, Figure 5b, and Figure 6b.

Only the highest stress values obtained in the experimental and numerical methods were compared as long as they were in the same region of the component. Table 3 shows the comparison of the results obtained by finite elements with those of photoelasticity. According to this table and with the other results, the following observations can be made:(a)The results of photoelasticity are in agreement with those obtained by finite elements since the tensions were concentrated in the same regions;(b)The difference of the principal stresses of the finite element analysis was obtained from the maximum points for each stress (which are not necessarily coincident) in a three-dimensional model. For the photoelastic analyses, this difference of the principal stresses was obtained for the same point and in two-dimensional figures. This may have influenced the percentage difference between the numerical and experimental results presented in Table 5;(c)During the heating of the resin model in the stress freezing method, the Poisson’s coefficient may have increased and influenced the difference between the results;(d)The SLA-Clear epoxy resin was suitable for photoelastic analysis and the stress freezing method because the resin is transparent and provides good optical response generating well-defined isochromatic fringes. In addition to retaining the stress distribution after the thermal process;(e)The cage had no photoelastic fringe, so it is assumed that the function of the cage is to keep the rolling elements equally spaced;(f)Considering the difference between experimental and numerical results, it can be concluded that photoelasticity can assist during design when numerical results are far from the correct result. Checking the results using the technique described in this paper before constructing a prototype can be more economical and safer than a design error prototype;(g)The printing of photoelastic models by the stereolithographic process is very useful and makes experimental analysis much easier since the time spent producing the models can be used to perform more constructive research tasks;(h)The possibility of making complex models by the stereolithographic process may increase interest in photoelastic technique and create interest in developing more suitable materials for photoelastic analysis.

## 5. Conclusions

In this research, a model of a tapered roller bearing in stereolithography with transparent resin was developed. This model was analyzed using the stress freezing method, which allowed the analysis of several components in contact. Rapid prototyping was adequate for photoelastic analysis. The advantages of the proposed method over other experimental methods are: measure the stress field interior to the model, analyze several components in contact and analyze components that have a complex shape. In addition, the method can be used to validate or verify results obtained in numerical stress analysis techniques such as the finite element analysis, providing more reliability and safety in prototype design.

The method proposed in this research increases the range of applications of three-dimensional photoelasticity and demonstrates that it is possible to analyze a complex model (composed of several parts). Printing the bearing using stereolithography makes the production of the photoelastic model viable, since producing by the conventional process would be very complicated and time consuming. The stress freezing method was successful with the SLA-Clear resin allowing to check of the stress state in the model after loading. This is a broad and promising field of research. Much research has been conducted on the properties of transparent resins when subjected to stress loading and subsequent slicing. The advance of rapid prototyping, especially 3D printing, should contribute to more advances in photoelasticity. Much research has been conducted on the properties of transparent resins when subjected to stress loading and subsequent slicing. This is a broad and promising field of research.

## Figures and Tables

**Figure 1 materials-12-03427-f001:**
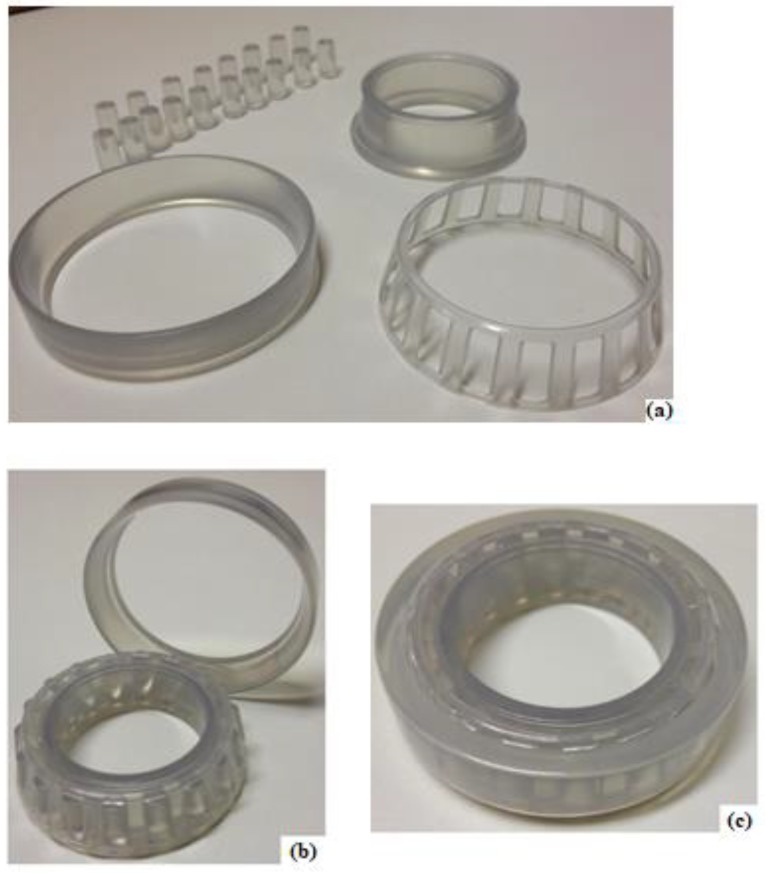
(**a**) Sample of tapered roller bearing made in stereolithography (SLA)-separate components. Sample of tapered roller bearing made in stereolithography (SLA): (**b**) separate outer ring, and (**c**) mounted bearing.

**Figure 2 materials-12-03427-f002:**
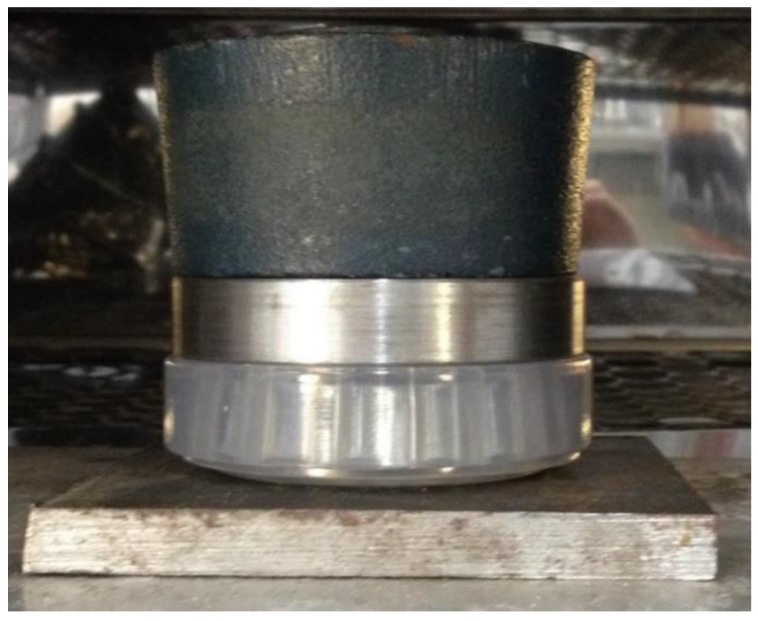
Freezing of stresses-prototyped model in stereolithography.

**Figure 3 materials-12-03427-f003:**
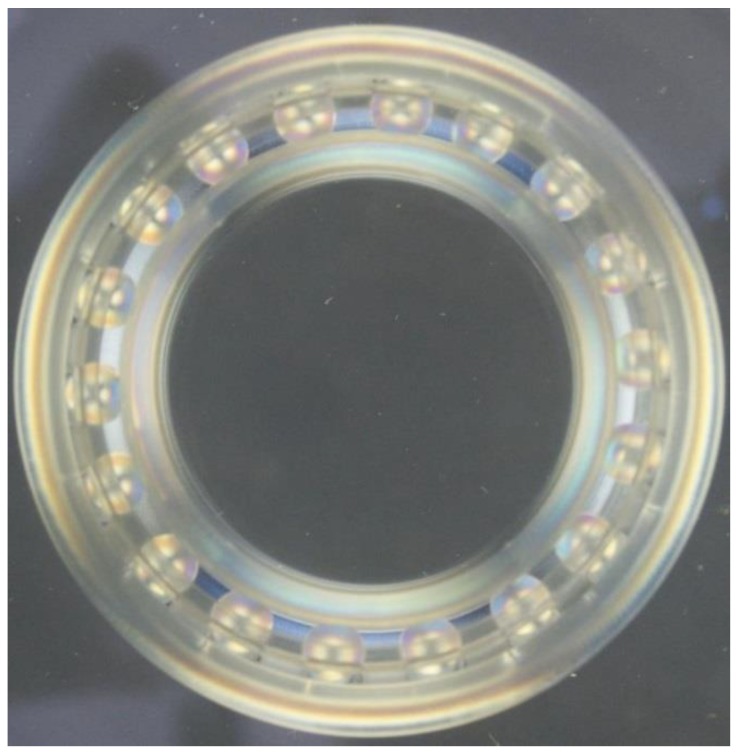
Photoelastic image of the tapered roller bearing model subjected to axial load.

**Figure 4 materials-12-03427-f004:**
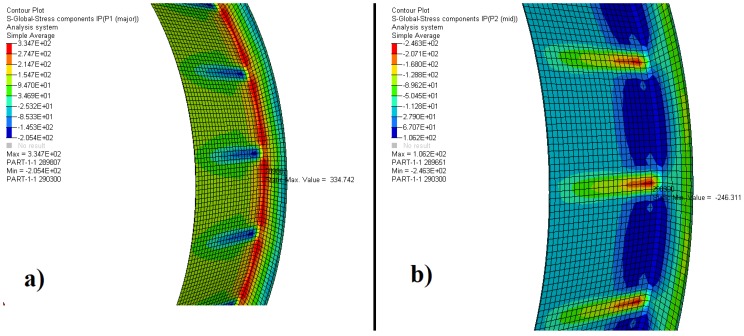
Distribution of the (**a**) first and (**b**) second principal stress-outer ring.

**Figure 5 materials-12-03427-f005:**
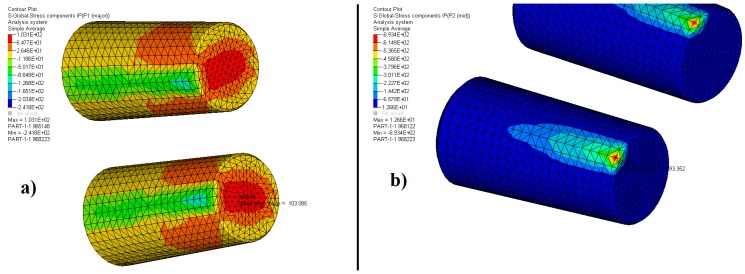
Distribution of the (**a**) first and (**b**) second principal stress-tapered rollers.

**Figure 6 materials-12-03427-f006:**
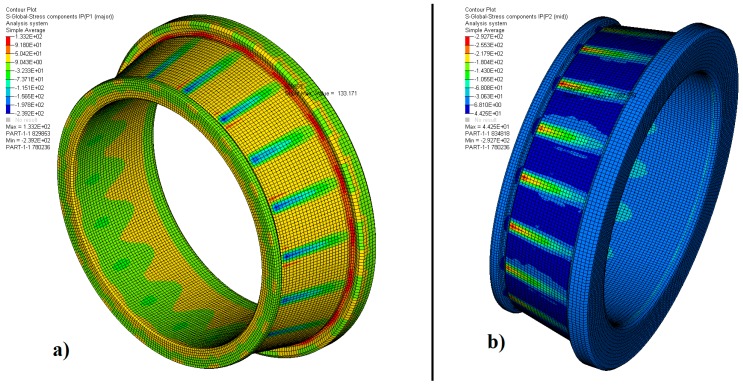
Distribution of the (**a**) first and (**b**) second principal stress-inner ring.

**Table 1 materials-12-03427-t001:** Properties of the epoxy resin stereolithography (SLA)-Clear.

Property	
Hardness (Shore)	86 D
Specific mass	1.21 g/cm^3^
Stretching	5–13% (at break)
Deflection temperature at 0.4551 MPa	53–55 °C
Deflection temperature at 1.8202 MPa	48–50 °C
Tensile strength	58–68 MPa
Modulus of elasticity	2.69–3.10 GPa
Bending strength	87–101 MPa
Modulus of flexion	2.70–3.00 MPa
Glass transition temperature	58 °C

**Table 2 materials-12-03427-t002:** Main dimensions of the bearing studied.

Description	Value	Unit
Initial contact angle	11.60	°
Final contact angle	14.67	°
Flange angle	76.87	°
Angle of conical roller	3.07	°
Length of conical roller	14.63	mm
Smallest diameter of conical roller	6.73	mm
Largest conical roller diameter	7.48	mm
Width of outer ring	15.50	mm
Width of inner ring	20.00	mm
Bearing outer diameter	75.00	mm
Bearing inner diameter	45.00	mm
Initial diameter-inner race	50.34	mm
Final diameter-outer race	71.16	mm
Effective bearing center	16.00	mm

**Table 3 materials-12-03427-t003:** Principal stresses in tapered roller bearing-photoelasticity.

Difference of the Principal Stresses	Cone Roller	Outer Ring	Inner Ring
(σ1–σ2) [MPa]	4.03	3.34	4.00

**Table 4 materials-12-03427-t004:** Principal stresses in tapered roller bearing-finite element analysis.

Principal Stresses	Cone Roller	Outer Ring	Inner Ring
σ1 [MPa]	103.1	334.7	133.2
σ2 [MPa]	−693.4	−246.3	−292.7
(σ1–σ2) [MPa]	796.5	581.0	425.9

**Table 5 materials-12-03427-t005:** Comparison of results.

Difference of the Principal Stresses	Cone Roller	Outer Ring	Inner Ring
σp = (σ1–σ2) [MPa]	796.5	581.0	425.9
σm = (σ1–σ2) [MPa]	4.03	3.34	4.00
σ = σp/σm	197.6	174.0	105.7
F = Fp/Fm	160	160	160
E = |σ−F|/F	23.5%	8.8%	33.9%

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
