# Peer review of "Analysis of Stresses in a Tapered Roller Bearing Using Three-Dimensional Photoelasticity and Stereolithography"

_materials, 2019, doi:10.3390/ma12203427_

Round 1

Reviewer 1 Report

The results presented in the article seem interesting and useful. However, the presentation of the results is very short and meager. The advantages of this method are not convincingly presented. I believe that the article can be published after revision, especially the sections “Results and Discussion” and “Conclusion”

Figures 1 and 2 are superfluous and do not add the information stated in the text. Figures 3 and 4 can be combined into one drawing in the following sequence: figure 3; figure 4 b; figure 4 a. The “Results and Discussion” section is too short and does not clarify the main results and conclusions. The FEA abbreviation in table 3 is not defined. The results presented in table 3 do not follow from the corresponding text and are not clear. The conclusion is too general and does not answer the question - what are the advantages of the proposed method compared to known methods, for example, the finite element method? What additional information can be obtained using the proposed method and what is its value?

Author Response

Response to reviewer 1 comments

Point 1: The results presented in the article seem interesting and useful. However, the presentation of the results is very short and meager.

Response 1: We added more figures and tables to detail the results.

Point 2: The advantages of this method are not convincingly presented. I believe that the article can be published after revision, especially the sections “Results and Discussion” and “Conclusion”.

Response 2: The sections “Results and Discussion” and “Conclusion” were expanded and revised.

Point 3: Figures 1 and 2 are superfluous and do not add the information stated in the text.

Response 3: Figures 1 and 2 were removed.

Point 4: Figures 3 and 4 can be combined into one drawing in the following sequence: figure 3; figure 4 b; figure 4 a.

Response 4: Figures 3 and 4 were combined.

Point 5: The “Results and Discussion” section is too short and does not clarify the main results and conclusions.

Response 5: The “results and discussion” section was expanded to clarify and explain the results.

Point 6: The FEA abbreviation in table 3 is not defined.

Response 6: The meaning was added.

Point 7: The results presented in table 3 do not follow from the corresponding text and are not clear.

Response 7: This table was  modified and two paragraphs were added in the "methodology" section to explain the values in the table.

Point 8: The conclusion is too general and does not answer the question - what are the advantages of the proposed method compared to known methods, for example, the finite element method? What additional information can be obtained using the proposed method and what is its value?

Response 8: “Conclusion” section was expanded to explain the importance of the results obtained. Among experimental techniques, only photoelasticity can measure the stress field interior to the model using stress freezing method. Unfortunately the numerical methods still are more attractive due to low cost, good precision and simplicity. But photoelasticity can still be used to compare results and have more security in the design phase.

Note: I made a mistake and put wrong data in table 5, the current data is the correct one.

Reviewer 2 Report

The authors of the manuscript present interesting and important research. I strongly agree with Authors that the preparation of the prototype can be very expensive and difficult, thus, it is a need to develop numerical methods that help to produce prototype and predict the properties of the final product.

Some comments are below.

Introduction: too little information about novelty of the presented research.  Figure 1 is not clear enough. Are the properties of the epoxy resin SLA-Clear presented in table 1 from the literature/given by producer of this resin? What is the material from which the tapered roller bearing are made? How does the model of the numerical method reflect the real product? As far as I understand the polymer for the presented experiments where selected according to its transparency and possibility to able photoelasticity and stereolithography.

Author Response

Response to reviewer 2 comments

Point 1: Introduction: too little information about novelty of the presented research.

Response 1: The following paragraph was added in introduction:

“The bearing resin model is identical to the original device made of steel and no part was removed from the bearing to simplify experiments, which is common when the analyzed component has complex shape. Therefore, there is no change in geometry that can influence the results. Research on three-dimensional photoelasticity and photoelastic models produced by stereolithography is rare. Thus, this research seeks to contribute to the development of three-dimensional photoelastic analysis by showing that producing models by stereolithography may be the only viable alternative for some components.”

Point 2: Figure 1 is not clear enough.

Response 2: Figure 1 was removed.

Point 3: Are the properties of the epoxy resin SLA-Clear presented in table 1 from the literature/given by producer of this resin?

Response 3: The properties were given by producer.

Point 4: What is the material from which the tapered roller bearing are made?

Response 4: The material is ASTM 52100 bearing steel.

Point 5: How does the model of the numerical method reflect the real product?

Response 5: We consider the application of this bearing in manual gearboxes in first gear, in this case the bearing is subjected to axial effort of 2.04 kN. To analyze the model, the axial load due to the maximum torque in first gear was applied. For even distribution of this load, rigid elements were created, distributed in 360º and connected to the bearing by the inner ring. The union of these rigid elements occurs at the bearing centerline, where the axial load mentioned above was applied. All steel data provided by the bearing manufacturer was included in the software to perform the analysis.

Point 6: As far as I understand the polymer for the presented experiments where selected according to its transparency and possibility to able photoelasticity and stereolithography.

Response 6: Yes that is correct.

Note: I made a mistake and put wrong data in table 5, the current data is the correct one.

Round 2

Reviewer 1 Report

There is no limit to perfection.

Of course, the text can be improved.

I believe that at this stage the results can be published to familiarize the relevant community.

Reviewer 2 Report

Thank you for response to all my comments in the detailed way. Now I can recommend this manuscript to be published in Materials.